# Production of Primary Metabolites by *Rhizopus stolonifer*, Causal Agent of Almond Hull Rot Disease

**DOI:** 10.3390/molecules27217199

**Published:** 2022-10-24

**Authors:** Anjali Zaveri, Jacqueline Edwards, Simone Rochfort

**Affiliations:** 1School of Applied Systems Biology, La Trobe University, Bundoora, VIC 3083, Australia; 2Agriculture Victoria Research, 5 Ring Road, Bundoora, VIC 3083, Australia

**Keywords:** *Rhizopus* species, carbon and nitrogen sources, almond hull composition, fumaric acid, lactic acid

## Abstract

Species in the fungal genus *Rhizopus* are able to convert simple sugars into primary metabolites such as fumaric acid, lactic acid, citric acid, and, to a lesser extent, malic acid in the presence of specific carbon and nitrogen sources. This ability has been linked to plant pathogenicity. *Rhizopus stolonifer* causes hull rot disease in almonds, symptoms of which have been previously associated with the fungus’s production of fumaric acid. Six isolates of *R. stolonifer* taken from infected almond hulls were grown in artificial media amended with one of four carbon sources (glucose, fructose, sucrose, and xylose) and two nitrogen sources (asparagine and ammonium sulphate) chosen based on almond hull composition and used in industry. Proton nuclear magnetic resonance (^1^H NMR)–based metabolomics identified that *R*. *stolonifer* could metabolise glucose, fructose, sucrose, and to a lesser extent xylose, and both nitrogen sources, to produce three metabolites, i.e., fumaric acid, lactic acid, and ethanol, under in vitro conditions. Sugar metabolisation and acid production were significantly influenced by sugar source and isolates, with five isolates depleting glucose most rapidly, followed by fructose, sucrose, and then xylose. The maximum amounts of metabolites were produced when glucose was the carbon source, with fumaric acid produced in higher amounts than lactic acid. Isolate 19A–0069, however, preferred sucrose as the carbon source, and Isolate 19A–0030 produced higher amounts of lactic acid than fumaric acid. This is the first report, to our knowledge, of *R. stolonifer* producing lactic acid in preference to fumaric acid. Additionally, *R. stolonifer* isolate 19–0030 was inoculated into Nonpareil almond fruit on trees grown under high– and low–nitrogen and water treatments, and hull compositions of infected and uninfected fruit were analysed using ^1^H NMR–based metabolomics. Glucose and asparagine content of uninfected hulls was influenced by the nitrogen and water treatments provided to the trees, being higher in the high–nitrogen and water treatments. In infected hulls, glucose and fructose were significantly reduced but not sucrose or xylose. Large amounts of both fumaric and lactic acid were produced, particularly under high–nitrogen treatments. Moreover, almond shoots placed in dilute solutions of fumaric acid or lactic acid developed leaf symptoms very similar to the ‘strike’ symptoms seen in hull rot disease in the field, suggesting both acids are involved in causing disease.

## 1. Introduction

*Rhizopus* species are zygomycete fungi well known for their ability to produce primary metabolites such as fumaric acid, lactic acid, citric acid, ethanol, and, to a lesser extent, malic acid in the presence of certain carbon and nitrogen sources [1,2]. Studies have found that various carbon sources such as glucose, sucrose, xylose, fructose, mannose, cellobiose, fatty acids, and starch can be utilised by *Rhizopus* species to produce organic acids [2,3,4,5,6,7,8]. For example, *R. oryzae* has been reported to produce 0.88 g lactic acid and 1.3 g fumaric acid per g D–glucose consumed [9,10]. Due to this ability, *Rhizopus* species have been extensively studied for the commercial production of organic acids (i.e., fumaric acid and lactic acid) and enzymes [5,7,8,11,12,13,14]. Additionally, a nitrogen source is required for organic acid production. Inorganic nitrogen, such as urea and ammonium sulphate [(NH_4_)_2_SO_4_], are utilised in the process of organic acid production by *Rhizopus* species in bio–fermentation industries [13]. *Rhizopus* species are also able to metabolise amino acids as their nitrogen source [15,16,17], although this is not economically feasible for large–scale production [18].

Aside from industrial importance, the ability of certain fungi to produce organic acids has been linked to plant pathogenicity. Certain plant pathogenic fungi produce organic acids to acidify the environment as a strategy to damage host tissue [19,20,21,22,23]. Necrotrophic fungi such as *Sclerotinia sclerotiorum* and *Botrytis cinerea* secrete oxalic acid as their major virulence factor in causing plant disease [19,21], while *Aspergillus niger* and *Penicillium* species secrete combinations of gluconic and citric acid [24,25,26]. The most striking example of the effect of organic acid toxicity on a plant host is the hull rot disease of almonds. In California, hull rot is mostly attributed to the fungus *R. stolonifer* and occasionally other *Rhizopus* and *Monilinia* species, *Aspergillus niger,* and *Neoscytalidium dimidiatum* [27,28]. In a significant study in California, Mirocha et al. [29] demonstrated that *R. stolonifer* colonised the split almond hull and produced toxins that caused twig dieback without the fungus colonising the affected leaf and twig tissue. They identified the candidate toxin as fumaric acid, with malic, citric, and tartaric acid as metabolic derivatives [29].

Hull rot disease has two distinct symptoms. First is the rotting of the hull after hull split occurs as a result of colonisation by *R*. *stolonifer,* where the rotted hull, together with the kernel, remains attached to shoots that bear them, known as spurs. Second, the spur is killed, and twig dieback follows (termed a strike) because of acid metabolites produced by the fungus and translocated into the shoot [29,30]. The disease causes yield loss as repeated episodes of hull rot on the tree progressively kill more fruiting wood [30,31]. Sometimes disease symptoms are limited to the rotting of the hull and do not progress further into twig dieback. Hull rot is known as a ‘good growers’ disease’ because the disease is often worse in well–maintained orchards provided with plenty of water and nitrogen to promote growth and productivity [31,32,33,34,35]. In California, there have been several studies on the effect of varying rates of nitrogen and the scheduling and timing of irrigation on the incidence and severity of hull rot [32,35,36,37,38,39]. These studies have demonstrated that restricting water supply to the almond trees as a pre–harvest treatment significantly reduced hull rot [32,37,38]. Similarly, the percentage of infected hulls and the number of hull rot strikes per tree were positively correlated with increased rates of nitrogen application [36].

In order for fungi to produce organic acids, they require carbon and nitrogen sources from their environment [26,40]. Carbohydrates and amino acids are a part of the composition of almond hulls [41,42]. Various studies have identified that almond hulls contain fermentable sugars such as fructose, sucrose, and glucose, sugar alcohols such as sorbitol and inositol, as well as polysaccharides and other components such as amino acids, crude protein, soluble protein, fibres, and minerals [41,42,43]. In a recent study, we analysed the hull composition of the Australian almond fruit of Nonpareil and Carmel using proton nuclear magnetic resonance (^1^H NMR)–based metabolomics and showed that the hulls contained the sugars glucose, sucrose, fructose, and xylose, branch–chain amino acids, and asparagine [44]. Hulls from the most commonly grown variety Nonpareil, which is more susceptible to hull rot than the polliniser variety Carmel, had higher sugar and asparagine contents than Carmel hulls, and the hull composition changed when different nitrogen and water treatments were applied to the trees. Additionally, Nonpareil hulls had relatively high glucose and asparagine content when high rates of nitrogen and water (the industry standard treatment) were provided to the trees. However, restricted irrigation raised the sucrose component, restricted nitrogen raised the glucose component, and restricting both nitrogen and water raised the sucrose and asparagine components [44]. These changes in hull composition under different nitrogen and water scenarios have the potential to affect the growth of *R. stolonifer* and its metabolite production in hull rot disease. Therefore, in the current study, *R. stolonifer* was grown in culture media containing one of four sugars (glucose, sucrose, fructose, or xylose) as the carbon source and one of two nitrogen sources (ammonium sulphate or asparagine) to investigate its ability to produce primary metabolites under these conditions. Additionally, infected and uninfected Nonpareil almond hulls grown under four different water and nitrogen regimes in a commercial orchard were collected and analysed to explore the utilisation of sugars and asparagine in the host environment to produce fumaric acid or other metabolites.

After 53 years, this is the first time almond hull composition and *R. stolonifer* metabolite production in hull rot disease have been studied. The current study utilised an NMR metabolomics approach to investigate metabolite production of (a) *R. stolonifer* isolates collected from infected hulls with and without associated twig dieback symptoms, grown in vitro using carbon and nitrogen sources based on Nonpareil hull composition and used in the industry and (b) *R*. *stolonifer* in infected Nonpareil almond hulls grown under different nitrogen and water regimes in the orchard. The objectives were to determine which sugars were preferred as a carbon source by *R. stolonifer*, which sugars or asparagine are required for organic acid production by *R. stolonifer*, whether *R. stolonifer* can utilise asparagine as a nitrogen source, whether there is a difference in organic acid profiles produced by *R. stolonifer* isolates associated with and without twig dieback, and whether the different water and nitrogen conditions that trees were subjected to in the orchard influenced the production of organic acids in infected hulls.

## 2. Results

*R*. *stolonifer* isolates collected from six Australian almond orchards were grown in culture media with different carbon and nitrogen sources (three replicates). The four carbon sources investigated were the sugars found in Australian almond hull composition in our previous study, i.e., glucose, sucrose, fructose, and xylose [44]. The culture medium also contained asparagine or ammonium sulphate [NH_4_(SO_4_)_2_] as nitrogen sources. Ammonium sulphate [NH_4_(SO_4_)_2_] is commonly utilised as a nitrogen source in commercial applications of *Rhizopus* species [13]. The sugar and nitrogen metabolisation for each isolate were analysed by NMR. The NMR spectra of each isolate were examined by principal component analysis (PCA). The unsupervised method describes the variance between the samples. Although the data set is complex, with variation due to the isolate, nitrogen source, and days in culture, PCA of the NMR dataset showed clustering of samples based on the sugar (Figure 1 and Appendix A). This suggests that most spectral variations are due to the differences between individual sugars. Given that sugar was the major variable influencing the data, the NMR spectra for each isolate were then analysed separately to determine the sugar and nitrogen utilisation of each isolate in the production of primary metabolites.

### 2.1. In Vitro Sugar and Nitrogen Metabolization by Rhizopus stolonifer

Sugar and nitrogen metabolisation of *Rhizopus stolonifer* isolates were studied to determine their sugar and nitrogen preferences, as found in almond hull composition. To achieve this, the regions for carbohydrates in the NMR spectra from δ 4.60–4.68 ppm for glucose, δ 4.085–4.124 ppm for fructose, δ 5.36–5.48 ppm for sucrose, and δ 5.175–5.199 ppm for xylose were extracted from the spectra and summed for each isolate. There was significant overlap in the oxymethine and oxymethylene regions of the spectra that was due to the different sugars, but the regions chosen for each sugar were unique to that sugar. Therefore, this measure gave an approximation of relative sugar concentration in culture media for each isolate. There was a considerable variation between each sugar concentration according to days in culture, and there was also variation between the isolates. The differences between the sugar concentrations were statistically significant (*p* < 0.05) (Appendix A). The NMR spectra showed that five isolates, 19A–0008, 19A–0010, 19A–0030, 19A–0036, and 19A–0139, preferred glucose and fructose as a carbon source in the presence of asparagine as the nitrogen source (Figure 2). The concentrations of glucose and fructose quickly decreased in the culture media and were depleted after 4 days by isolates 19A–0010, 19A–0030, and 19A–0036 and 5 to 6 days by isolates 19A–0008 and 19A–0139. Sucrose was depleted after 5 days by isolates 19A–0010 and 19A–0036 and 6 days by 19A–008 and 19A–0139, while a small amount of sucrose was still present on the 10th day for isolate 19A–0030. Isolate 19A–0069 utilised sucrose more rapidly than the other sugars. For most isolates, xylose was metabolised preferentially to sucrose for the first few days of the experiment. There were two exceptions, isolate 19A–0069, which only slowly metabolised xylose, and isolate 19A–0139, where there was no significant difference between the metabolisation of xylose and sucrose.

Considering that NH_4_(SO_4_)_2_ is utilised in industrial bio–fermentation as the nitrogen source, we also analysed sugar consumption in the presence of NH_4_(SO_4_)_2_ as a control to determine the effect of nitrogen source on the sugar preference of *R. stolonifer*. As for asparagine, glucose and fructose were the preferred carbon sources for the five isolates 19A–0008, 19A–0010, 19A–0030, 19A–0036, and 19A–0139 in the presence of NH_4_(SO_4_)_2_ as it was in the presence of asparagine (Figure 3, Appendix A), although the sugar metabolisation by three isolates was slower with NH_4_(SO_4_)_2_. Glucose was depleted in the culture media after 8 days by isolates 19A–0008, 19A–0010, and 19A–0036, with NH_4_(SO_4_)_2_ compared with 4–5 days with asparagine, whereas with isolates 19A–0030 and 19A–0139, glucose was depleted in 4 days with both NH_4_(SO_4_)_2_ and asparagine. Sucrose and xylose were, again, the least preferentially metabolised sugars, and some quantity remained on the 10th day. Isolate 19–0069 differed from all other isolates as it again preferred sucrose followed by glucose and fructose.

### 2.2. Production of Primary Metabolites

Three metabolites, fumaric acid, lactic acid, and ethanol, could be clearly identified from the ^1^H NMR spectra (Appendix A). Upfield regions of spectra were dominated by ethanol, lactic acid, asparagine, and sugars. An expansion of the downfield region showed that all the isolates produced fumaric acid. The NMR spectra of fumaric acid, lactic acid, and ethanol were analysed by extracting the spectra and summing for each isolate in the same manner as for the sugars in Section 2.1. The following regions were extracted and summed: for fumaric acid, from δ 6.51 to 6.61 ppm (corresponding to the two olefinic methines of the acid); for lactic acid, from δ 1.30 to 1.38 ppm (corresponding to the three protons of the methyl moiety), and for ethanol, from δ 1.15 to 1.2 ppm (corresponding to the three protons of the methyl moiety). The spectral regions have few resonances, and those present are due to fumaric acid, lactic acid, and ethanol. Therefore, the results are an approximation of relative fumaric acid, lactic acid, and ethanol concentrations in each sample. There is a considerable variation in fumaric acid, lactic acid, and ethanol concentration between isolates and sugar sources but not between nitrogen sources (Figure 4, Figure 5, Figure 6, Figure 7, Figure 8 and Figure 9). Fumaric acid production by isolates 19A–0010, 19A–0036, and 19A–0139 was highest when glucose was the carbon source, followed by fructose (Figure 4 and Figure 7). Fumaric acid production in all isolates was lowest when sucrose and xylose were the carbon sources. Of all the isolates, 19A–0030 produced the least amount of fumaric acid on all of the sugars. All isolates produced lactic acid on all sugars and nitrogen sources (Figure 5 and Figure 7). In all isolates, the relative concentration of lactic acid produced was highest when glucose was the carbon source, followed by fructose, sucrose, and xylose. Isolate 19A–0030 produced the highest amount of lactic acid by almost two orders of magnitude. Ethanol was produced as a by–product in all cultures, along with fumaric acid and lactic acid (Figure 6 and Figure 9).

The differences in the overall amount of fumaric acid, lactic acid, and ethanol produced by each isolate were statistically significant (*p* < 0.05) (Appendix A). The effects of isolates, sugar, and days were all statistically significant (*p* < 0.05), but the effect of the nitrogen source was not significant. Therefore, the differences were due to variation between individual isolates, their preference for carbon source, and the rate at which they metabolised the sugars over the 10–day period. In general, all isolates demonstrated the ability to utilise the sugars and asparagine present in almond hulls to produce fumaric acid and lactic acid. There was some differentiation between isolates according to whether they were associated with strike symptoms or not. Isolates 19A–0010, 19A–0030, and 19A–0036 from infected hulls with strike symptoms took 3 to 4 days for sugar and asparagine metabolisation, whereas isolates 19A–0008, 19A–0069, and 19A–0139 from infected hulls without strike symptoms took 5 to 6 days.

### 2.3. Sugar and Nitrogen Metabolisation of Rhizopus stolonifer in Infected Hull Samples

Nonpareil almond fruit grown under two levels of irrigation (high and low) and two levels of nitrogen (high and low) were inoculated using isolate 19A–0030, and the hull metabolites were analysed by NMR. The NMR spectra of infected hulls were examined by Principal Component Analysis (PCA). The unsupervised method describes the variance between the samples. PCA of the NMR dataset showed the separation of samples based on the changes in the hull composition of the infection (Appendix A). As found in our earlier study, the hull composition, in general, contained simple sugars, including glucose, fructose, sucrose, and xylose, and the amino acids asparagine, alanine, valine, iso–leucine, and leucine [44]. In this study, the levels of sugar and nitrogen were investigated further to determine changes in the composition of the infected hull samples grown under the four nitrogen–irrigation regimes. The spectral regions for the analytes of interest were δ 4.60–4.68 ppm for glucose, δ 4.09–4.12 ppm for fructose, δ 5.36–5.48 ppm sucrose, δ 5.18–5.20 ppm for xylose, δ 2.82–2.88 ppm for asparagine, δ 6.51–6.61 ppm for fumaric acid, and δ 1.30–1.38 ppm for lactic acid. They were extracted and summed for the infected and control samples as described previously.

The NMR spectra showed that the extractable metabolome in infected hull samples was low in carbohydrates and asparagine compared with the control samples (Appendix A). Additionally, two metabolites, fumaric acid and lactic acid, were identified in the infected hulls (Appendix A). This suggests that the spectral variation observed was due to sugar and nitrogen metabolisation and metabolite production by *R. stolonifer* (Figure 10a,b). Sucrose, fructose, and glucose were significantly lower in infected hull samples compared with that of controls under all treatment conditions, with little difference in xylose levels between the infected and control hull samples (Figure 10a). The production of fumaric acid and lactic acid was influenced by the different nitrogen and water treatments that trees were subjected to in the orchards (Figure 10c). Fumaric acid and lactic acid production was higher under high–nitrogen treatments and reduced under low–nitrogen treatments. Lactic acid production was higher than fumaric acid production in infected hulls (1.5–fold difference in data when normalised).

Results of ANOVA—simultaneous component analysis showed that fixed effects (infection status and treatments) had the largest effect on the levels of the metabolites, although there was an interaction effect as well (Appendix A). In general, the infection status had the largest effect on the level of the metabolites, with the exception of glucose and asparagine, which were affected more by water and nitrogen levels, although the interaction effect was also high in these cases. There was more glucose and asparagine present in the hulls when the trees were given more nitrogen and more water (Figure 10a,b). There was little difference in levels of sucrose and xylose between the control and infected hulls (Figure 10a), possibly because glucose and fructose were not limiting and were preferentially metabolised. The levels of fumaric acid and lactic acid were high in all the infected samples but highest under high–nitrogen conditions. Overall, this analysis demonstrates that the nitrogen and irrigation treatments that are applied in the orchard affect the glucose and asparagine levels inside the hulls and that acid metabolite production in infected hulls is also modified under these conditions.

### 2.4. Toxicity of Fumaric Acid, Lactic Acid, and Ethanol to Almond Shoots

Toxicity analysis of fumaric acid, lactic acid, and ethanol on Nonpareil almond shoots showed that both fumaric and lactic acid were toxic on almond shoots, and both could play a role in causing hull rot disease symptoms (Figure 11 and Appendix A). Symptoms in leaf and shoots after 24 h and 72 h are shown in Appendix A. Almond shoots in 0.05 M fumaric acid solution showed curling, twisting, brittleness, and dark–brown discolouration of their mid–veins (Figure 11a and Appendix A). Necrosis was also visible in some leaf petioles. These symptoms are consistent with classic ‘strike’ symptoms. The shoots treated with 0.05 M lactic acid showed curling and brown discolouration of their mid–veins (Figure 11b and Appendix A). However, the brittleness of the leaf was not observed. Ethanol caused only a slight discolouration after 48 h (Figure 11c and Appendix A). Almond shoots were healthy in the control treatment (Figure 11d and Appendix A).

## 3. Discussion

This study has shown that *R*. *stolonifer* isolates collected from infected almond samples utilise sugars and nitrogen found in almond hulls and produce the primary metabolites fumaric acid, lactic acid, and ethanol. Analysis of the sugar and nitrogen utilisation of *R*. *stolonifer* isolates grown in vitro revealed that the isolates can utilise glucose, fructose, sucrose, and xylose as carbon sources regardless of nitrogen source ((NH_4_)_2_SO_4_ or asparagine) provided, but the different isolates metabolised the sugars at different rates. This was investigated in planta when the hulls were infected with isolate 19A–0030, and the amino acid asparagine, the sugars glucose, fructose, and sucrose were depleted as the fumaric and lactic acid were produced. Therefore, asparagine present in the almond hull can be utilised as a nitrogen source by *R*. *stolonifer* for metabolite production. In planta, xylose was not utilised by the fungus and ethanol was not detected, probably due to its volatility. Each *R. stolonifer* isolate investigated was able to utilise the sugar and amino acid constituents of almond hulls for organic acid production, and the in vitro data were indicative of metabolisation in infected hulls in the orchard.

Differential carbohydrate utilisation depends on the *Rhizopus* species and strain and has been studied for the commercial production of organic acids. For instance, in the efficient production of lactic acid and citric acid, sucrose served as the carbohydrate for some strains of *Rhizopus* species [45,46]. *R. oryzae* strain GY18 produced 97.5 g L^−1^ L–lactic acid from 120 g L^−1^ sucrose and was also able to utilise glucose and xylose to produce high amounts of lactic acid [45]. In 2015, Nwokoro investigated citric acid production from orange peel waste using a strain of *R. stolonifer* [46]. The basal medium for citric acid production contained sucrose, lactose, and maltose at three concentrations of 5, 10, and 15% (*w*/*v*), respectively. When sucrose, maltose, and lactose were all provided at 15% (*w*/*v*) concentration, sucrose yielded the highest amount of citric acid at 33.7 g L^−1^, followed by maltose at 26.8 g L^−1^, and lactose at 22.6 g L^−1^ [46]. However, there are some *Rhizopus* species and strains that poorly metabolise sucrose compared with glucose [6,8]. The effect of carbon source on fumaric acid production for a *Rhizopus* sp. strain isolated from brown rice was investigated [6]. The carbon sources included in this study were glucose, sucrose, fructose, glycerol, lactose, maltose, starch, and galactose. The results showed that maximum fumaric acid production of 16 g L^−1^ was obtained by metabolising glucose and fructose, followed by maltose, starch, galactose, and lastly, glycerol [6]. In contrast, no fumaric acid production was observed using sucrose as the carbon source.

In our study, we found that sugar metabolisation and acid production of *R*. *stolonifer* was significantly influenced by sugar sources and the specific isolate. Analysis of six individual isolates showed that *R*. *stolonifer* could metabolise glucose, fructose, sucrose and, to a lesser extent, xylose in in vitro conditions. There was also a noticeable difference in the rate of in vitro sugar metabolisation between isolates that produced strike symptoms in planta and those that did not. The glucose and fructose concentrations were reduced rapidly and depleted after 4–5 days in vitro for isolates capable of producing strike symptoms (19A–0010, 19A–0030, and 19A–0036), whereas they took 5–6 days to deplete sucrose and xylose. Isolates 19A–0008, 19A–0069, and 19A–0139 that were not associated with strike symptoms took 5–6 days to deplete the glucose and fructose, and the sucrose and xylose were not fully consumed after 10 days in vitro.

Fumaric acid production was highest when the isolates utilised glucose and fructose, followed by sucrose and xylose. Significant differences were observed in lactic acid production by the isolates, with isolate 19A–0030 producing high amounts of lactic acid compared with other isolates. While all isolates produced fumaric acid, five isolates, 19A–0010, 19A–0036, 19A–0008, 19A–0069, and 19A–0139, did so rapidly in the presence of glucose and fructose, while the isolate 19A–0030 produced fumaric acid quite late (8–10 days) by utilising sucrose as carbon source. Moreover, all isolates were able to produce lactic acid, but isolate 19A–0030 produced 100 times more lactic acid than the other five isolates. There are no reports on the production of lactic acid by *R. stolonifer* in in vitro conditions in the bio–fermentation industry. All isolates produced ethanol, which is a by–product of fumaric and lactic acid production [8,9,10,47]. Isolate 19A–0030 also produced higher lactic acid than fumaric acid in the infected hulls (1.5–fold higher when data were normalised).

Along with specific carbohydrate sources, *Rhizopus* species also require a nitrogen source to produce organic acids. Inorganic nitrogen sources such as ammonium sulphate ((NH_4_)_2_SO_4_) and urea are used in bio–fermentation industries to increase the metabolic activities of fungal cells over fungal growth [13]. *Rhizopus* species can metabolise amino acids as nitrogen sources [16,17,48], but this can be economically challenging for large–scale production of organic acids [18]. In our previous experiments, we identified high concentrations of asparagine in almond hulls and wondered whether asparagine was a potential nitrogen source for *R*. *stolonifer* [44]. As a part of our current study, we found that all six *R*. *stolonifer* isolates were able to utilise asparagine when provided in the culture medium. There were no significant differences in metabolism or acid production based on nitrogen source, indicating that *R. stolonifer* can efficiently utilise asparagine as a nitrogen source. This was supported by the in planta data, where asparagine, along with glucose, sucrose, and fructose, was reduced in the infected hull samples, providing evidence that *R. stolonifer* in the infected hulls had utilised these to produce fumaric and lactic acid. Therefore, the sugar and nitrogen components of almond hulls provide substrates for *R. stolonifer* to produce fumaric acid and lactic acid in hull rot disease.

The production of fumaric and lactic acids by *R. stolonifer* is particularly of interest as Mirocha and Wilson (1961) demonstrated the disease development cycle of hull rot whereby *R. stolonifer* colonised inside the hull split but did not invade and colonise the affected leaf and twig tissue [30]. They identified the involvement of toxin production by *R. stolonifer* that caused the leaf blighting and twig dieback using paper chromatography [29], but they did not report any traces of lactic acid. *R. stolonifer* was grown in the water extract of almond hulls along with anion and cation exchange resins. Paper chromatographic analysis of the cation exchange resin aliquot showed high concentrations of fumaric acid and trace amounts of succinic acid [29]. These acids were not found in the anion–exchanged aliquot. An equal amount of citric and malic acid appeared in all test solutions [29]. They have tested the toxicity of 0.05 M potassium fumarate and sodium succinate on almond shoots. Shoots in fumaric acid showed symptoms that resemble the blighted shoots in the field. However, shoots in succinic acid did not show any symptoms. They did not determine whether the fumaric acid metabolic derivatives malic and citric were also toxic and could cause hull rot symptoms [29]. In our in vitro study, we identified three metabolites, i.e., fumaric acid, lactic acid, and ethanol, produced by *R. stolonifer* isolates using an NMR metabolomics approach. We did not identify succinic, citric, or malic acids. In the analysis of infected hulls, only fumaric and lactic acid were identified, but not ethanol (which is volatile and may have evaporated or been removed in sample processing, e.g., freeze–drying). Significantly, lactic acid and ethanol production was not found in previous hull rot studies. Our toxicity analysis of the metabolites showed that both fumaric acid and lactic acid (0.05 M) were toxic to almond shoots and induced symptoms resembling that of hull rot strikes, although leaf brittleness was observed only with fumaric acid. Ethanol did not induce symptoms. For better comparison, we used concentrations of the fumaric acid and lactic acid similar to those of the Mirocha and Wilson study as the only study of hull rot disease. This suggests that both fumaric acid and lactic acid production by *R. stolonifer* can potentially be involved in causing strike symptoms.

In California, field studies demonstrated that strike symptoms caused by *R. stolonifer* included hull rot, leaf blighting on the adjacent spur, discolouration of the mid and lateral veins of the leaves, and leaf curling which later became brittle, leading to spur death [32,35,36,37,38]. These studies have found that nitrogen and water treatments provided to almond trees influenced the incidence and severity of hull rot in Californian orchards. Restricting water supply to trees before harvest or using regulated deficit irrigation significantly reduced hull rot [32,37,38]. Similarly, reducing the nitrogen supply reduced hull rot severity in orchards [35,36]. In our previous and present studies, we found that restricted water and nitrogen treatments applied in the orchard influenced the sugar and nitrogen ratios in the hull [44]. In our previous study, we found that in Nonpareil, high–nitrogen–high water treatment (the control) had relatively high glucose and asparagine content. High–nitrogen–low–water treatment increased the sucrose component, low–nitrogen–high–water treatment increased the glucose component, and low–nitrogen–low–water treatment increased the sucrose and asparagine components. Similarly, significant differences were observed in this current study between the sugar and nitrogen levels of uninfected hulls from the different nitrogen –water treatments. In the current experiment, there was little effect on sucrose in control samples. Asparagine was reduced under low nitrogen –low water compared with high–nitrogen–high–water (*p* = 0.01) controls. High–nitrogen–high–water treatment had high glucose and asparagine levels in uninfected (control) almond hulls. High–nitrogen–low–water treatment increased the asparagine content, low–nitrogen–high–water treatment increased the glucose component, and low–nitrogen–low–water treatment increased the sucrose content. In infected hull samples, glucose, fructose, and asparagine levels were reduced under high–nitrogen–high–water treatment, whereas only minor differences were observed in sucrose and xylose levels under any treatment in both infected and uninfected hull samples. This demonstrates that *R. stolonifer* will preferentially utilise glucose and fructose over sucrose and xylose, and asparagine is a preferred nitrogen source. Fumaric acid and lactic acid were both produced abundantly in infected samples but were highest in the high–nitrogen treatments because of the presence of glucose and asparagine. This suggests that manipulating the water and nitrogen applied in the orchard subsequently affects the amount of glucose and asparagine available in the hulls for *R. stolonifer*, which, in turn, affects the amount of fumaric acid and lactic acid produced inside the infected hulls.

In summary, this current study has demonstrated that sugar and nitrogen sources influence the metabolite production of *R. stolonifer* isolates collected from infected almonds. *R. stolonifer* preferred glucose as a carbon source, followed by fructose, sucrose, and, to a lesser extent, xylose, and was able to utilise both ammonium sulphate and asparagine as the nitrogen source in vitro. Three metabolites, i.e., fumaric acid, lactic acid and ethanol, were produced by *R. stolonifer* utilising these sugar and nitrogen sources in vitro. Metabolite production varied with isolate and sugar, with the simple sugars glucose and fructose generally preferred. One isolate, 19A–0030, preferentially produced lactic acid rather than fumaric acid and preferred sucrose, unlike the five other isolates, which produced fumaric acid and preferred glucose and fructose. Infected hull analysis showed glucose, followed by fructose and asparagine, were reduced compared with uninfected hulls, but sucrose and xylose were not affected, supporting the in vitro results and demonstrating that *R. stolonifer* prefers glucose and asparagine for growth and metabolite production. Glucose and asparagine were reduced when nitrogen and water were restricted in the orchard. There were two metabolites, fumaric acid and lactic acid, identified in the infected hulls. The changes in glucose and asparagine levels in the hulls under different nitrogen and water treatments corresponded to changes in metabolite production, with higher fumaric and lactic acid production in the high–nitrogen treatments. To conclude, *R. stolonifer* isolated from infected almond hulls produce fumaric acid and lactic acid by utilising the glucose and asparagine present. Trees that are provided with ample nitrogen and water have more glucose and asparagine in the hulls, such that acid metabolite production by *R. stolonifer* is high. While previous studies identified fumaric acid as the candidate toxin causing hull rot strikes [29], in the current study, we have identified both fumaric acid and lactic acid as candidate toxins for hull rot strike symptoms in the field. Future research should examine more closely the role of both fumaric and lactic acids in symptom development and whether that can be manipulated with in–field orchard management.

## 4. Materials and Methods

### 4.1. Site Descriptions of Fungal Isolates and Culture Conditions of In Vitro Metabolite Analysis

Six *Rhizopus stolonifer* isolates were used in this study and were isolated from infected almond fruits collected from various almond orchards during an industry–wide disease survey in 2018 (Table 1). Three isolates came from hull rot associated with twig dieback (strike) and three from hull rot without associated dieback (without strike), and they were chosen to provide some geographic spread. All the isolates were grown on potato dextrose agar (PDA) plates at 25 °C for 10 days. Spore suspensions were prepared as mentioned in 4.3 inoculation preparation, adjusted to a concentration of 10^5^ spores/mL using a haemocytometer.

#### 4.1.1. Culture Medium

For in vitro culture conditions and culture media were prepared based on the method utilised by Kowalczyk et al. [8] with modifications in the quantity of spore suspension, carbohydrate, and nitrogen sources. Each medium contained a single carbohydrate source (glucose, fructose, sucrose, or xylose) 40 g/L, KH_2_PO_4_ g/L, and MgSO_4_.7H_2_O 0.5 g/L, a single nitrogen source ((NH_4_)_2_SO_4_ or asparagine) 1.4 g/L, CaCl_2_ 0.3 g/L, and 0.5 mL of microelements solution (FeSO_4_.7 H_2_O 5 g/L, MnSO_4_.H_2_O 1.96 g/L, and ZnSO_4_ 1.66 g/L), and distilled water added to make a final volume of 1 L. All media were sterilised by autoclaving at 121 °C for 20 min. The carbohydrates were prepared and sterilised separately, then added to the medium before inoculations. CaCO_3_ (1 g) was added to glass tubes and sterilised in a drying oven at 150 °C for 3 h to avoid contamination when added separately to each flask before inoculations. All experiments were performed in triplicate per isolate. Carbohydrate and asparagine combinations were chosen to represent almond hull compositions from our previous study [44] to ascertain which were preferred by *R. stolonifer* for metabolite production. The asparagine was replaced with (NH_4_)_2_SO_4_ to provide controls based on bio–fermentation industry protocols.

#### 4.1.2. Culture Conditions

Spore suspension (1 mL) was inoculated into 150 mL jars containing 40 mL of culture medium. The cultures were grown at 25 °C, 200 rpm in rotary shakers for 10 days. 2% g/v CaCO_3_ was added to cultures when the pH dropped below 4.5 (measured after every 24 h of culture) to return to pH 4.5.

#### 4.1.3. NMR Sample Preparation

Every 24 h during the 10 days, 1 mL was aliquoted per fungal culture medium and filtered through Whatman filter paper into a 1.5 mL Eppendorf tube. Samples were stored at 4° C prior to extraction. A 550 µL aliquot was taken from each 1 mL sample and added to new 1.5 mL Eppendorf tubes, and 50 µL of D_2_O solution was added prior to a 30 s vortex (Ratek vortex mixer, VM1, Boronia, VIC, Australia) followed by 5 min sonication (SoniClean, 250TD, Stepney, SA, Australia). An aliquot of 500 µL was transferred into 5 mm NMR tubes.

### 4.2. Site Description, Fungal Isolates, and Culture Conditions of In Vitro Metabolite Analysis

The experimental site was located on a commercial orchard at Lindsay Point, VIC, Australia, planted in 2005 with Nonpareil as the main variety and Carmel as a polliniser. The orchard rows were oriented east–west with row and tree spacings of 7.2 m × 4 m, respectively. The experimental layout was a replicated block design with six replicates per treatment. The experimental unit was three trees per replicate plot.

The treatments consisted of two levels of irrigation (high and low) and two levels of nitrogen (high and low). High nitrogen was the commercial practice recommended by the Almond Board of Australia (ABA) for the major mineral elements N:P:K 320:40:400. The low–nitrogen treatment was applied as 56% of the high–nitrogen treatment (N:P:K 180:40:400). The high–water treatment was irrigation at 100% ETc estimated using the ABA–recommended crop factors. The low water treatment was applied as 70% ETc throughout the growing season as a sustained deficit. The irrigation and nitrogen treatments were established five years prior to the start of the experiment.

### 4.3. Inoculum Preparation

*Rhizopus stolonifer* isolate 19–0030 (Table 1) was grown on potato dextrose agar (PDA) for 10 days at 20 to 22 °C under 24 h light conditions. Spores were washed from culture plates with sterile water and filtered through Miracloth^®^ to remove mycelial fragments and clumped spores. The spore suspension was counted with a haemocytometer and adjusted to a concentration of 10^5^ spores/mL.

### 4.4. Inoculation

Nonpareil fruits were inoculated on 21 January 2020. Three Nonpareil almond trees that were bearing a sufficient number of healthy fruits that had reached hull split were chosen per treatment. Thirty fruits were selected across the three trees, and each one was inoculated with 1 mL *R. stolonifer* spore suspension. A further 30 healthy fruit were inoculated with 1 mL of water as a control treatment. Each fruit was covered with a paper bag for 24 h to promote infection establishment.

### 4.5. Sample Collection

Ten days after inoculation, the fruits were harvested. The collected fruits were separated into hull, shell, and kernel. All the samples were snap–frozen in liquid nitrogen and stored in paper bags at −80 °C for at least 24 h before being freeze–dried in an ALPHA 1–4 LD freeze dryer (Martin Christ, Osterode am Harz, Germany) at −53 °C and 0.030 m Bar pressure for 48 h. Freeze–dried samples were stored at 4 °C.

### 4.6. Infected Almond Hull Sample Preparation for NMR

The freeze–dried hull samples were processed further for hull composition analysis. Ten infected and ten control hull samples per replicate from three replicates per treatment were used, resulting in 30 infected and 30 control samples per treatment. Further details of sample preparation and metabolite extraction protocol used in this study can be found in [44]. In brief, ground almond hulls (50 mg) were extracted with 1 mL D_2_O solution (5 mM 4,4–dimethyl–4–silapentane–1–sulfonic acid (DSS–d_6_), and 550 µL was transferred into 5 mm NMR tubes.

### 4.7. ^1^H NMR Data Acquisition and Preprocessing of Fungal Metabolites and Hull Composition

The data acquisition and preprocessing steps were as described in [44,49]. Standard 1D NMR spectra were obtained on a Bruker 700 MHz Avance^TM^ III NMR instrument equipped with a cryoprobe and SampleJet automatic sample changer with cooling (Bruker Biospin, Rheinstetten, Germany). All 1D NMR spectra were carried out using Bruker noesypr1d pulse sequence over −4 to 14 ppm spectral range by suppression of water resonance by presaturation. Acquisition parameters were as follows: spectral width, 11.08 ppm; acquisition time, 2.11 s per scan; time domain points, 32 K; the number of scans, 128 and 8 dummy scans. A line broadening of 0.3 Hz was applied to all spectra prior to Fourier transformation. A total of 1537 spectra were manually phased, baseline corrected in Topspin 4.0 (Bruker Biospin, Rheinstetten, Germany), and referenced to the DSS (non–deuterated) at δ 0.00 ppm. Assigned peaks were identified using Chenomx NMR suite software v.8.6 (Chenomx Inc., Edmonton, AB, Canada)

Data preprocessing was performed in MatLab (R2019a, Mathworks, Natick, MA, USA). Spectra were imported as a matrix of signal intensities using the ProMetab_v1_1 script [50]. Spectral preprocessing involved (1) deletion of the residual water peak region (δ 4.88–4.80 ppm), (2) normalisation to the total signal area (area = 1), (3) deletion of DSS (δ 0.4–0.60 ppm) peak region, (4) baseline adjustment using automatic weighted least squares (order = 2), and (5) mean centring.

### 4.8. Statistical Analysis

Multivariate statistical analyses were performed using the PLS Toolbox v. 8.9.2 (Eigenvector Research Inc., Manson, WA, USA). Preliminary data analyses of in vitro fungal metabolites and hull composition were performed using unsupervised PCA. Examination of PC1 vs PC2 scores plot showed asparagine samples from NH_4_(SO_4_)_2_ samples and infected hull samples from healthy hull samples outside the 95% confidence level ellipse (Appendix A).

Specific bins were summed to estimate individual metabolite levels in in vitro samples and infected hull samples: For fumaric acid, the spectral region from δ 6.51–6.61 ppm was extracted and summed; for lactic acid, the region from δ 1.30–1.38 ppm was used; for asparagine, the region from δ 2.82–2.88 ppm was used; for carbohydrates glucose, fructose, sucrose, and xylose, the regions from δ 4.60–4.68 ppm, δ 4.085–4.124 ppm, δ 5.36–5.48 ppm, and δ 5.175–5.199 ppm were used, respectively. These values were used to explore the influences of fixed effects in in vitro fungal metabolite samples (nitrogen, sugar, time, samples), and infected hull samples (treatment, replicates, and sample no.) on spectra were investigated using ANOVA simultaneous component analysis (ASCA) with following variables: 100 permutations, data autoscaled, remove centre points off, 2– and 3–way interactions calculated [50]. These areas were also used to calculate mean and standard deviation and to plot line graphs of metabolite levels in in vitro samples and infected hull samples.

One–way ANOVA with multiple comparison tests were carried out in MatLab for metabolite variables to analyse the significance of the in vitro sugar utilisation per day by each isolate and to assess differences between the infected hull and healthy hull composition from various nitrogen and water treatment combination applied to trees.

### 4.9. Toxicity Analysis of Metabolites Produced by R. stolonifer

For in vitro toxicity analysis, a 0.05 M solution of fumaric acid, lactic acid, or ethanol was prepared in 250 mL Erlenmeyer flasks containing 100 mL sterile water. A control treatment contained 100 mL of sterile water. Flasks were covered with parafilm. A Nonpareil almond shoot was placed into each of the flasks. All experiments were performed in triplicate per treatment. Leaf and twig symptoms were observed every 24 h for 72 h.

## Figures and Tables

**Figure 1 molecules-27-07199-f001:**
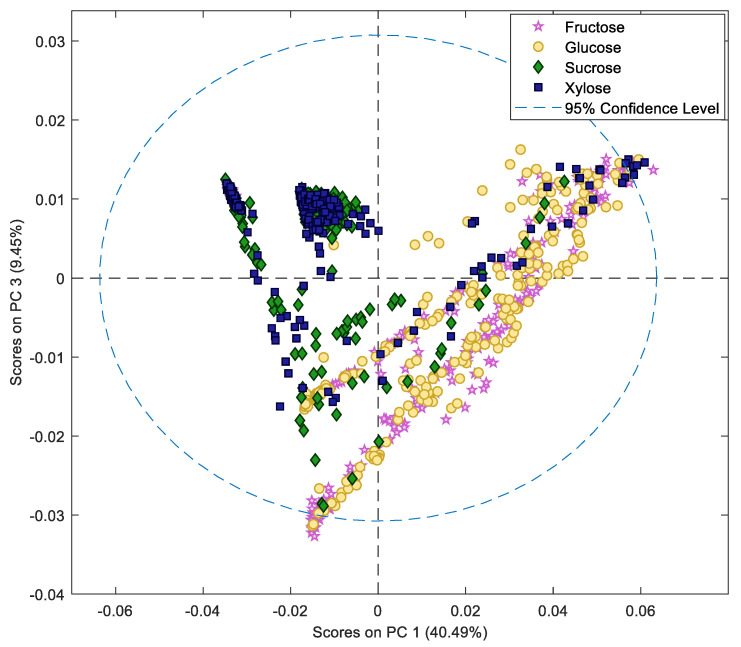
PCA score of ^1^H NMR data of aqueous extracts of in vitro *Rhizopus stolonifer* cultures. The PCA score plot shows the separation of samples based on sugar sources (glucose, fructose, sucrose, and xylose) regardless of isolates and nitrogen sources.

**Figure 2 molecules-27-07199-f002:**
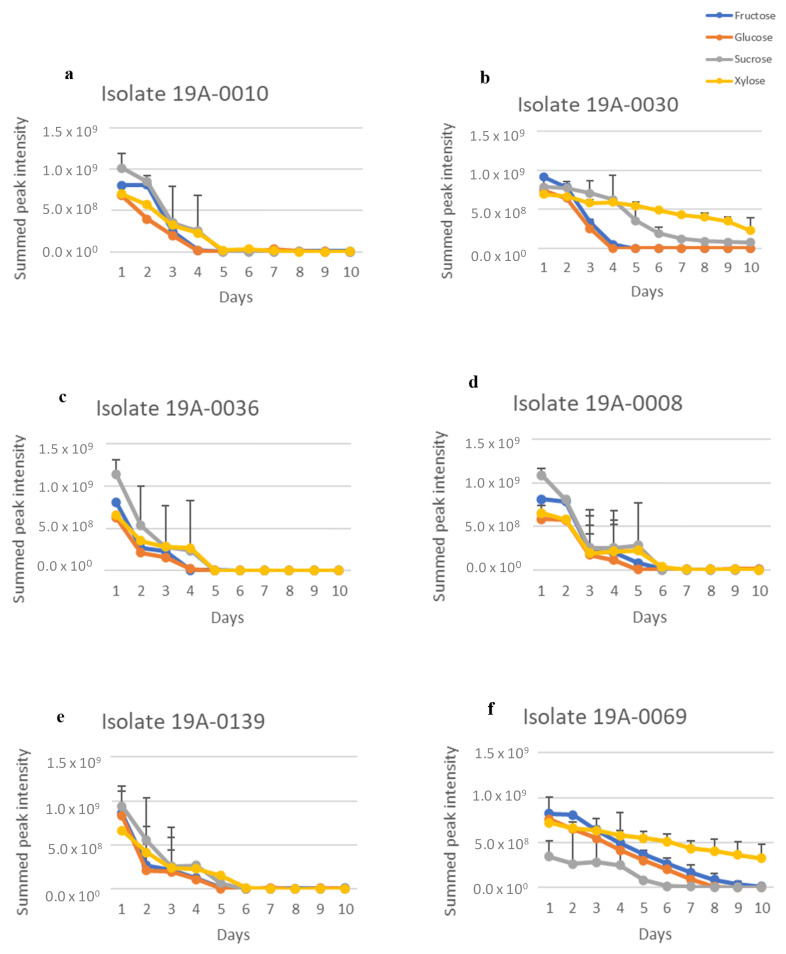
Sugar metabolisation by the different *Rhizopus stolonifer* isolates when asparagine was the nitrogen source. Data presented are the relative concentration of sugar (as determined from NMR) with standard deviation (error bars), *n* = 3. (**a**) Isolate 19A–0010, (**b**) isolate 19A–0030, (**c**) isolate 19A–0036, (**d**) isolate 19A–0008, (**e**) isolate 19A–0139, (**f**) isolate 19A–0069.

**Figure 3 molecules-27-07199-f003:**
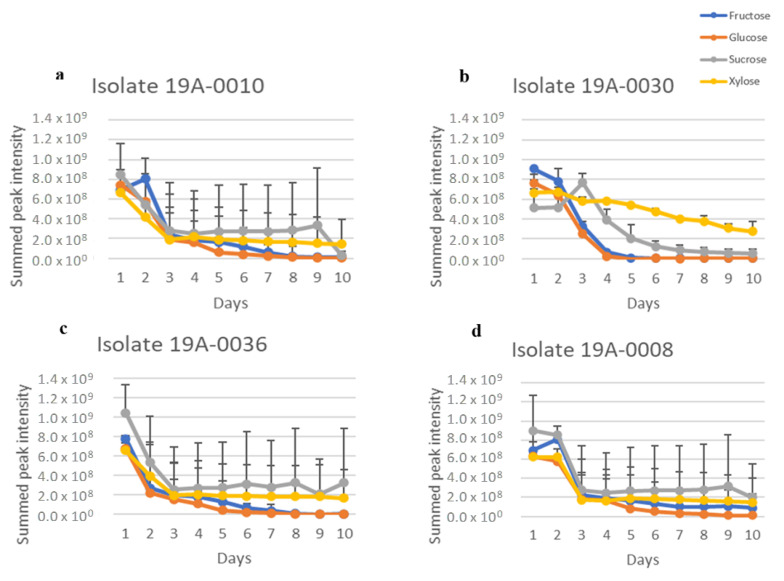
Sugar metabolisation of *R. stolonifer* isolates in the presence of NH_4_(SO_4_)_2_ as the nitrogen source. Data presented are the relative concentration of sugar (as determined from NMR) with standard deviation (error bars), *n* = 3. (**a**) Isolate 19A–0010, (**b**) isolate 19A–0030, (**c**) isolate 19A–0036, (**d**) isolate 19A–0008, (**e**) isolate 19A–0139, (**f**) isolate 19A–0069.

**Figure 4 molecules-27-07199-f004:**
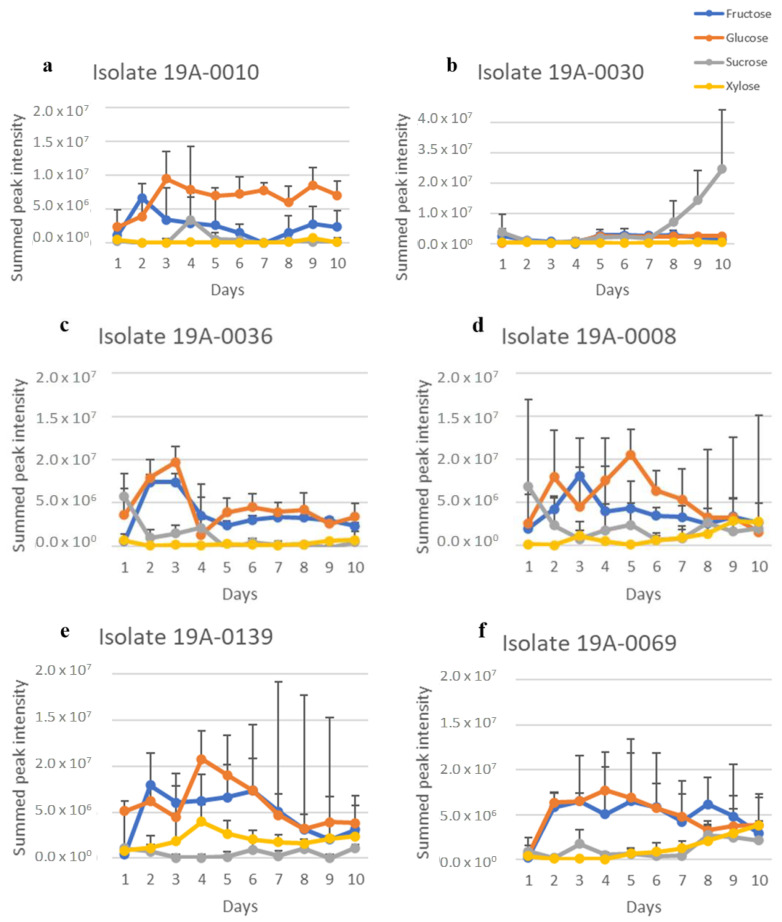
Fumaric acid production of *R. stolonifer* isolates in the presence of glucose, fructose, sucrose, or xylose, along with asparagine as the nitrogen source. Data presented are the relative concentration of fumaric acid (as determined from NMR) with standard deviation (error bars), *n* = 3. (**a**) Isolate 19A–0010, (**b**) isolate 19A–0030, (**c**) isolate 19A–0036, (**d**) isolate 19A–0008, (**e**) isolate 19A–0139, (**f**) isolate 19A–0069.

**Figure 5 molecules-27-07199-f005:**
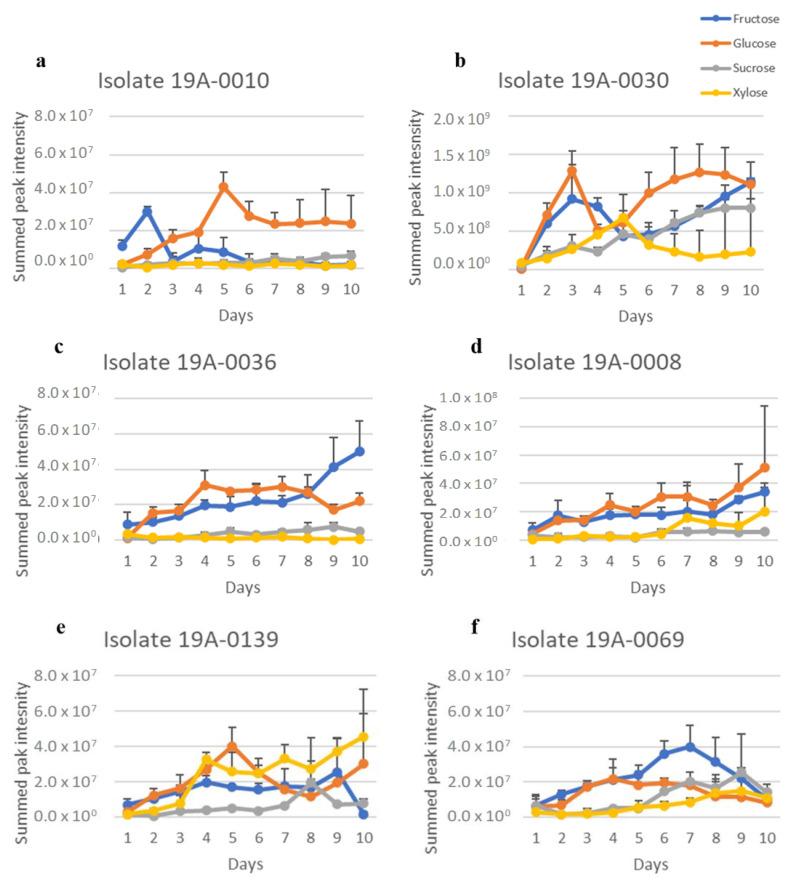
Lactic acid production of *R. stolonifer* isolates in the presence of glucose, fructose, sucrose, or xylose, along with asparagine as a nitrogen source. Data presented are the relative concentration of lactic acid (as determined from NMR) with standard deviation (error bars), *n* = 3. (**a**) Isolate 19A–0010, (**b**) isolate 19A–0030, (**c**) isolate 19A–0036, (**d**) isolate 19A–0008, (**e**) isolate 19A–0139, (**f**) isolate 19A–0069.

**Figure 6 molecules-27-07199-f006:**
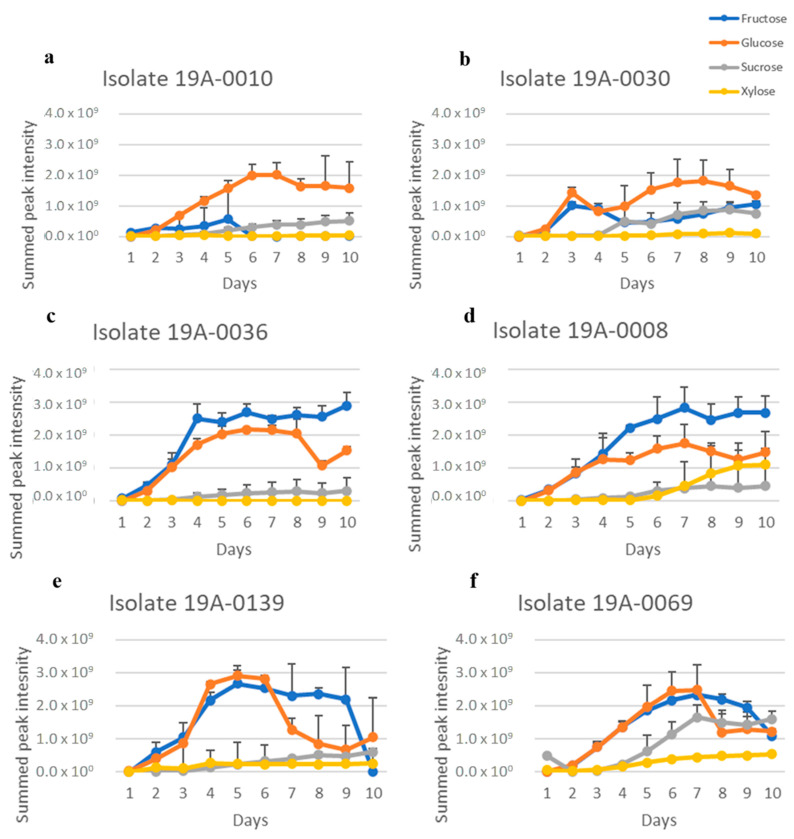
Ethanol production of *R. stolonifer* isolates in the presence of glucose, fructose, sucrose, or xylose, along with asparagine as a nitrogen source. Data presented are the relative concentration of ethanol (as determined from NMR) with standard deviation (error bars), *n* = 3. (**a**) Isolate 19A–0010, (**b**) isolate 19A–0030, (**c**) isolate 19A–0036, (**d**) isolate 19A–0008, (**e**) isolate 19A–0139, (**f**) isolate 19A–0069.

**Figure 7 molecules-27-07199-f007:**
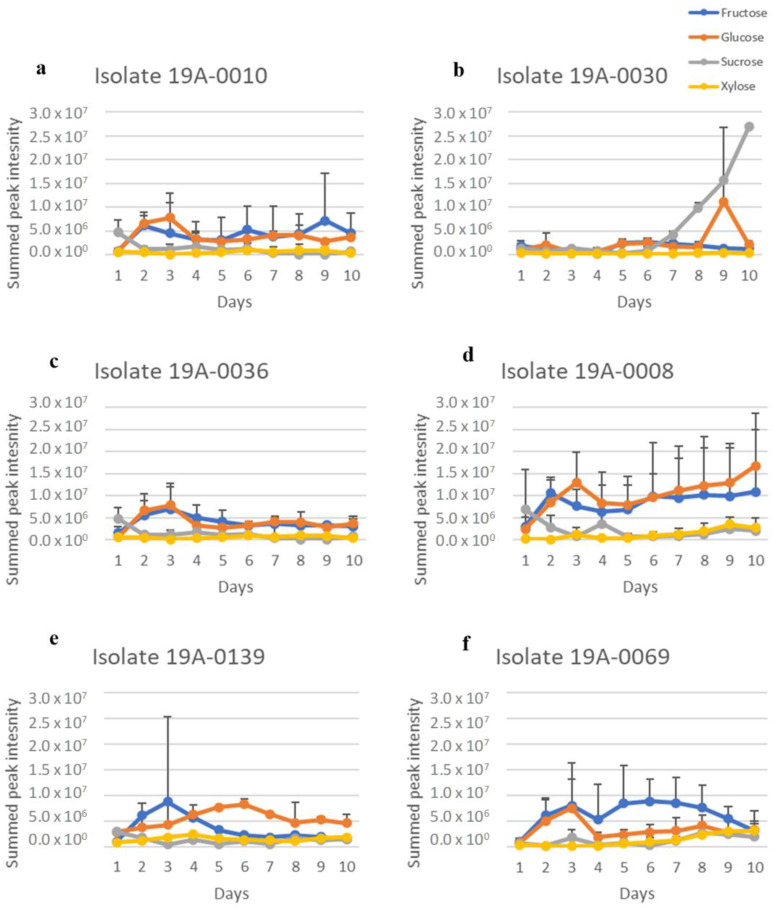
Fumaric acid production of *R. stolonifer* isolates in the presence of glucose, fructose, sucrose, or xylose, along with NH_4_(SO_4_)_2_ as a nitrogen source. Data presented are the relative concentration of fumaric acid (as determined from NMR) with standard deviation (error bars), *n* = 3. (**a**) Isolate 19A–0010, (**b**) isolate 19A–0030, (**c**) isolate 19A–0036, (**d**) isolate 19A–0008, (**e**) isolate 19A–0139, (**f**) isolate 19A–0069.

**Figure 8 molecules-27-07199-f008:**
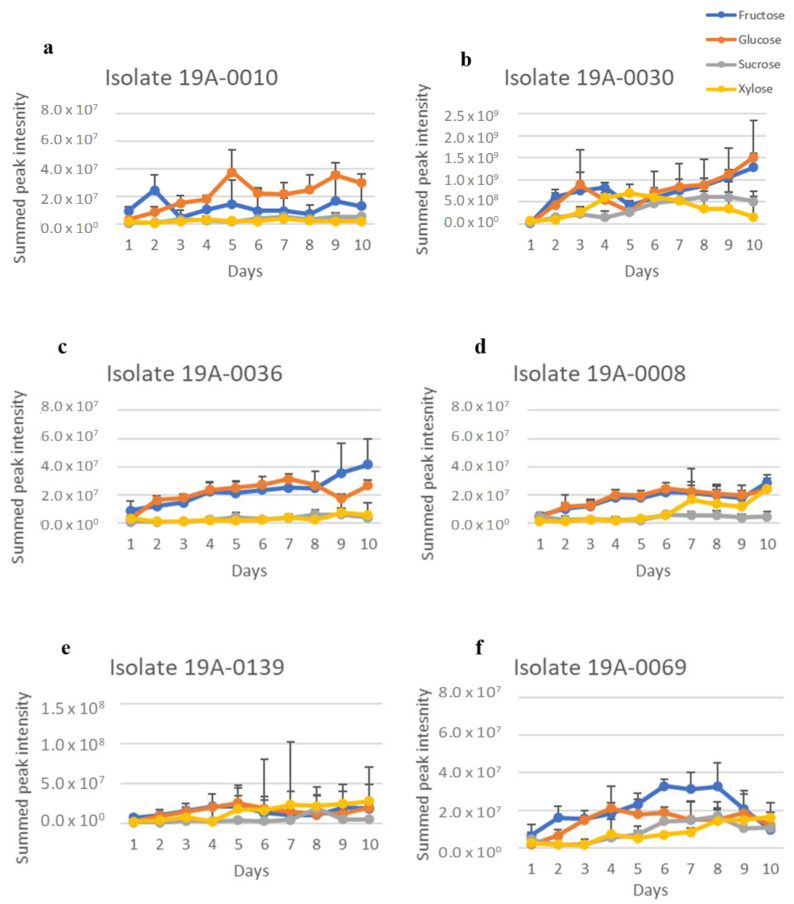
Lactic acid production of *R. stolonifer* isolates in the presence of glucose, fructose, sucrose, or xylose, along with NH_4_(SO_4_)_2_ as a nitrogen source. Data presented are the relative concentration of lactic acid (as determined from NMR) with standard deviation (error bars) *n* = 3. (**a**) Isolate 19A–0010, (**b**) isolate 19A–0030, (**c**) isolate 19A–0036, (**d**) isolate 19A–0008, (**e**) isolate 19A–0139, (**f**) isolate 19A–0069.

**Figure 9 molecules-27-07199-f009:**
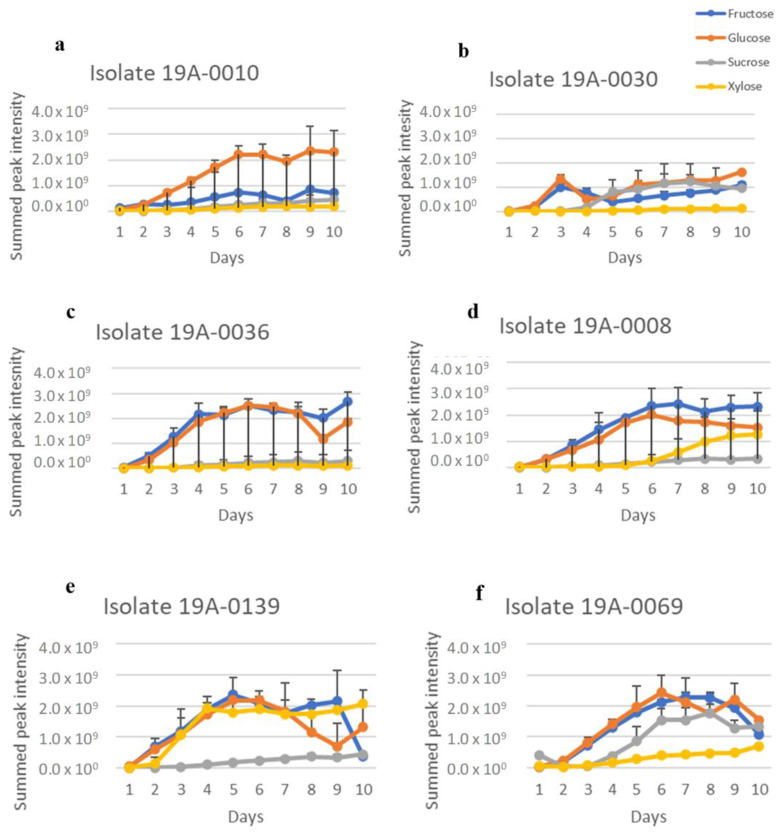
Ethanol production of *R. stolonifer* isolates in the presence of glucose, fructose, sucrose, or xylose, along with NH_4_(SO_4_)_2_ as a nitrogen source. Data presented are the relative concentration of ethanol (as determined from NMR) with standard deviation (error bars), *n* = 3. (**a**) Isolate 19A–0010, (**b**) isolate 19A–0030, (**c**) isolate 19A–0036, (**d**) isolate 19A–0008, (**e**) isolate 19A–0139, (**f**) isolate 19A–0069.

**Figure 10 molecules-27-07199-f010:**
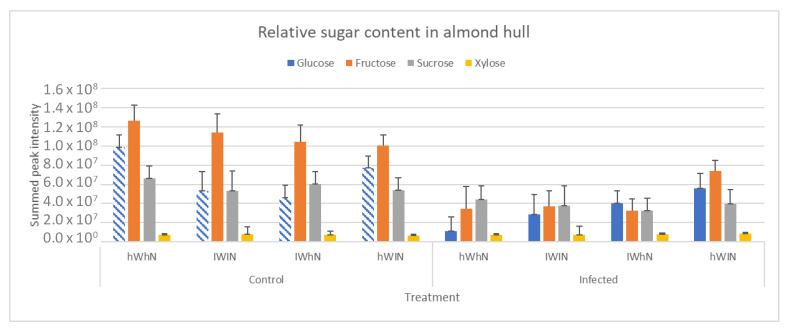
Relative sugar and metabolite content in the control and infected (isolate 19A–0030) hull samples with standard deviation. (**a**) Glucose, fructose, sucrose, and xylose (**b**) asparagine, (**c**) fumaric acid and lactic acid. Data presented are the relative concentration of sugars, fumaric acid, and lactic acid (as determined from NMR) with standard deviation (error bars), *n* = 30. Nitrogen and water treatments: hWhN—high–water–high–nitrogen, lWlN—low–water–low–nitrogen, lWhN—low–water–high nitrogen, hWlN—high–water –ow–nitrogen.

**Figure 11 molecules-27-07199-f011:**
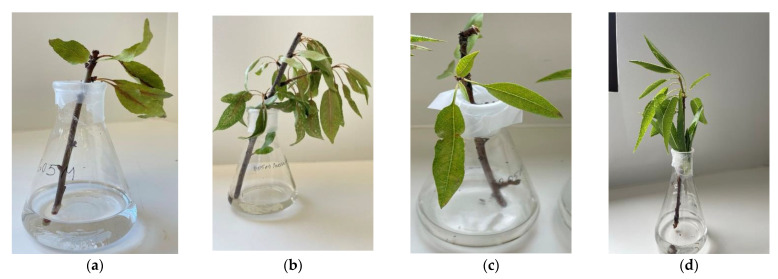
Metabolite symptoms in almond shoots after 48 h: (**a**) fumaric acid, (**b**) lactic acid, (**c**) ethanol, (**d**) control. The experiment was repeated in triplicate, and a representative image is shown here.

**Table 1 molecules-27-07199-t001:** *Rhizopus stolonifer* isolates used in this study, the almond variety and location that they came from, and the type of disease symptom they were associated with.

Isolate	Location	Variety *	Strike or without Strike
19A–0010	Merbein, VIC	NP	Yes
19A–0030	Cullullera, VIC	NP	Yes
19A–0036	Mildura, VIC	NP	Yes
19A–0008	Nangiloc, VIC	NP	No
19A–0069	Coleambally, NSW	P	No
19A–00139	Bennerton, VIC	C	No

* NP—Nonpareil, P—Price, C—Carmel.

## Data Availability

The data presented in this study are available on reasonable request from the corresponding author.

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
