# Peer review of "Production of Primary Metabolites by Rhizopus stolonifer, Causal Agent of Almond Hull Rot Disease"

_molecules, 2022, doi:10.3390/molecules27217199_

Round 1

Reviewer 1 Report

This article reports a chemical analysis of almond hull rot disease caused by selected isolates of Rhizopus stolonifer. Their quantitative analysis is based on NMR signals of both C or N sources (several sugars or ammonium sulfate and asparagine) and metabolites (several acids and ethanol). They found out that sugar metabolization and acid production are significantly influenced by sugar sources and isolates. They also succeeded in detecting symptoms of disease, which demonstrates that fumaric and lactic acids are involved in the cause of disease. The article presents all the necessary data and methods so that readers can pick up perspectives of the field. Therefore, the reviewer would recommend publication in the journal, if the authors consider the following minor points.

(1) About Figures 2-9 in page 5-13, the reviewer had a hard time to follow what the authors want to show with these similar-looking plots that continue through several pages. The reviewer would suggest the authors select representative data to clearly show what to be picked up. The rest of data can be put into the supporting materials so that readers can carefully follow those data one by one when they want. Please consider that kind of arrangement to make the paper fit in shape.

(2) About Figure 10 in page 14-17, plotting style is unclear and too lengthy. The reviewer suggests these 7 figures be put into 2 figures. For the problem of clearness, C/N sources and metabolites should be clearly separate. The present style is confusing, because all the graphs are independently plotted and readers would have to check labels and values one by one, which is really tiring. The reviewer believes bars for C/N sources (at least for sugars) can be put into the same graph, which may make comparison easy. The same rearrangement may be helpful for metabolites.

(3) In line 146 of page 4, the reviewer believes that the quantitative analysis of 1H NMR peaks is based on the integration (or area of signals) of appropriate signals. Exactly saying, the height of NMR signal is not completely parallel to the existence of those protons, but the area of signal should be. The authors should describe it exactly in here.

(4) There are some typos as follows: 1H NMR in line 140 (1 in superscript), 19A-008 in line 158 (0008), Isolate 19-A0030 in Figure 2b (19A-0030), Isolate 19-0030 in Figure 3 (19A-0030), NH4(SO4)2 in Figure 3 legend (4, 4, and 2 in subscript), 4.085-4.124 in line 302 (4.09-4.12), 5.175-5.199 in line 302 (5.18-5.20), 19-0030 in line 412 (19A-0030), 19-0036 in line 414 (19A-0036), "resemble to" in line 449 (resemble), 1H NMR in line 686 (1 in superscript).

(5) There are many cases where "fumaric acid and lactic acid", which are a bit busy when read out loud. Please consider replacement with "fumaric and lactic acids" unless the authors stick to the present description.

Author Response

10 October 2022

Manuscript ID: molecules-1936507

Dear Editor and reviewers

We greatly appreciate the thorough and thoughtful comments provided on our submitted article. We have revised our manuscript according to the reviewer’s comments and suggestions and we believe that as a result it has significantly improved. The corresponding changes and refinements made in the revised paper are a revised manuscript with track changes in it, revised supplementary data and a document summarising our response to each of the reviewer’s comments.

We thank the editors for their assistance and look forwarding to hearing from you in due course.

Best regards

Anjali Zaveri

PhD Candidate  

School of Applied Systems Biology | La Trobe University

Agriculture Victoria Research | Department of Jobs, Precincts, and Resources 
Agribio, Centre for AgriBioscience

5 Ring Road, La Trobe University, Bundoora, Victoria 3083, Australia 
T: 03 9032 7316 | M: 0420 724 373| anjali.zaveri@agriculture.vic.gov.au

Reviewer 2 Report

Six isolates of R. stolonifer have been analyzed by NMR for their production of multiple fermentation products, especially fumaric acid. The data are believable but my chief comment is that I found the data presentation less than compelling and often confusing. The results consist of a series of Figures, each of which has an axis of "summed peak intensity" versus 10 days, with no indication of which peaks were being summed for which metabolite or nutrient, or what the NMR data actually represented in terms of metabolite concentration. Most of the Figures went from 0 to E+08 to E+09, but the fumarate went from 0 to E+06 and E+07. Does that mean there is far more fumarate or less fumarate? It is difficult to make comparisons because each figure is taken up by comparisons among the 6 fungi. What is needed, urgently needed, is a Table or three with the relevant values being compared in the same Table, with the same concentration scale (mg/ml or uM or whatever) so that you can emphasize for the reader which differences you think are important. I often felt that the figures presented ought to be in the supplementary while I wanted to see a table you had put in the supplementary, like Table S2. Other points:

1/ You need to decide if this paper is intended for the industrial production of fumarate, the role of fumarate in a successful plant pathogen, or just a comparison of 6 fungi originally obtained from infected almond hulls. If you want to emphasize plant pathogenicity, you need some estimate of the pathogenicity of each fungal isolate, which you can then compare, for instance, to their maximum secretion of fumarate. If you are interested in the industrial production of fumarate, how do your production levels compare to for instance, Crueger & Crueger 1989 Table 8.1 giving a 65% conversion of glucose to fumarate in 3 days by Rhizopus growing at 33C. If instead you really want to compare these 6 isolates of R. stolonifer, you have to show me how they are interesting. Do they have a complete TCA cycle? Why are they carrying out a fermentation when grown aerobically? Do they give altered metabolites if you alter their iron availability? How do you get fumarate but not succinate? At present, I would keep two fungi for presentation in the text and confine the other 4 to supplementary because the presence of all six only confuses any points you want to make. 

2/ Please do a better job of telling the reader why you use NMR for your analysis. Your metabolite extraction section (lines 562-568) only confused me. Are you analyzing intracellular or extracellular metabolites. I had assumed extracellular, but then you are extracting 550 uL of sample with 50 uL of D2O. Something is missing. Did you rotovap or otherwise dry the 550 uL sample? 

3/ Your defined growth medium from reference 8 (2018) is basically a glucose-asparagine-salts medium. You say you use Asn because is present in infected almond hulls. Present as free Asn in the metabolic pool or present in some polymerized form? Also, glucose-Asn defined media have been used for R. stolonifer for many years. The Van Etten lab published roughly 20 papers in the 1970s using glucose-Asparagine-magnesium sulfate-potassium phosphate (see for instance Experimental Mycology 5 189-192 1981) and that medium had assuredly been developed over 10 years previously.

4/ I was impressed with your data lines 182-186 showing that all the glucose was used up in 8 days with ammonium sulfate but in 4-5 days with Asn. A change in pH is one obvious explanation. I would expect the Asn culture to stay around pH 6 wile the AS culture would acidify greatly. You had to add daily doses of calcium carbonate to keep the pH >4.5. It would be nice if you could follow pH more consistently, to see how much time the culture was < pH 4 before getting its daily dose of calcium carbonate. Did you ever try to use ammonium tartrate instead of AS?? The point is that if you could rule out pH effects then you would be left with more interesting reasons why the Asn cultures used up glucose much faster than the AS cultures. 

5/ Table 1, what does strike or w/o strike mean??

6/ Could you convert your ethanol production values to % values please? That would be very helpful in interpreting or predicting whether that ethanol would be harmful to fungal growth, or likely to damage plant tissues.

Author Response

(The authors gave the same response as above.)

Round 2

Reviewer 2 Report

Yes, the paper is now suitable for publication. As a philosophical matter, I agree that authors are not obligated to follow all of their reviewer's suggestions. There is sufficient merit in the paper as now written. Finally, in case you hadn't already guessed, I studied central metabolism and the nutritional requirements of Rhizopus stolonifer in a Plant Pathology department 48-49 years ago.